# The Needs and Utilization of Long-Term Care Service Resources by Dementia Family Caregivers and the Affecting Factors

**DOI:** 10.3390/ijerph17166009

**Published:** 2020-08-18

**Authors:** Chia-Hui Chang, Yung Ming, Tsung-Hung Chang, Yea-Yin Yen, Shou-Jen Lan

**Affiliations:** 1Department of Nursing, Taichung Veterans General Hospital, Taichung 40705, Taiwan; cjhsnd@vghtc.gov.tw; 2Central Office of Administration, Antai Medical Corporation Antai Tian-Sheng Memorial Hospital, Pingtung County 92842, Taiwan; 3Central Office of Administration, Taichung Veterans General Hospital, Taichung 40705, Taiwan; amingo6268@gmail.com; 4Department of Oral Hygiene, College of Dental Medicine, Kaohsiung Medical University, Kaohsiung 80708, Taiwan; yyyen0302@gmail.com; 5Department of Medical Research, China Medical University Hospital, China Medical University, Taichung 40447, Taiwan; shoujenlan@gmail.com

**Keywords:** dementia, caregiver of patients with dementia, caregiver burden, BPSD, long-term care service resources

## Abstract

This study was to evaluate the utilization of long-term care service resources by caregivers of patients with dementia (PWD) and to determine affecting factors. In this cross-sectional study, a total of 100 dyads were enrolled and caregivers responded to the questionnaires. We found 40% of caregivers not using any care resources. Between those caregivers using and not-using care resources, we found differences (*p* < 0.05) in their health status and living conditions; the difference (*p* < 0.05) was also found in patients’ behavior and psychological symptoms of dementia (BPSD). The frequency of BPSD (OR = 1.045, *p* = 0.016, 95% CI = 1.001–1.083) and the living conditions (OR = 3.519, *p* = 0.007, 95% CI = 1.414–8.759) were related to their use of care resources. Particular BPSDs, such as anxiety or restlessness, throwaway food, aggressive behavior, tearing of clothes, and sexual harassment of patients were related to the caregivers’ use of care resources (*p* < 0.01). Health professionals have to evaluate the patients’ BPSD and identify the caregivers’ essential needs. Individualized medical care and BPSD-related care resources should be provided for patients and caregivers for taking off their care burden and improving patient care.

## 1. Introduction

The world’s population is rapidly aging. Aging-related decline in physical and cognitive functions, especially dementia, creates a growing healthcare burden. The 2019 report of Alzheimer’s Disease International (ADI) estimated 50 million people in 2018 have dementia worldwide. By 2050, the number likely will rise to 152 million. A new case of dementia occurs in the world every three seconds. The global cost of the disease is about USD one trillion a year, and the cost is estimated to double by 2030 [1]. In Taiwan, according to the demographic statistics in December 2018, 280,783 people aged 65 and above have dementia, and they accounted for 7.78% of the elderly population. The estimated prevalence rate of dementia aged 65 and above in Taiwan is 12.39% by 2065, amounting to 890,000 people, or every 5 in 100 [2].

The main problem of dementia is cognitive dysfunction and the resulting deterioration in memory, orientation, judgment, calculation, abstract thinking, attention, and language. With reference to Behavior and Psychological Syndrome Dementia (BPSD), the functional disorders include agitation, personality change, delusions or hallucinations, aggression, wandering, sleep disturbance, and repetitive behaviors. The progression of these symptoms requires not only economic and social impacts in caring people with dementia (PWD), but also has detrimental effects on the well-being of caregivers [1,2,3,4]. Related to this issue, studies have shown that the caregivers of PWD often lead to physical stress (suffering from hypertension, cardiac diseases), psychological stress (depression, anxiety), social effects (limited activity, unable to work), and financial burdens, even increasing their risk of death and becoming “invisible patients”, leading to higher costs of personal health [5,6,7,8,9,10]. These do even consider having patients in institutional care prematurely, or into a care setting unsuitable for PWD [11,12,13,14]. During the decline of physical and cognitive function of PWD, it is known that notable behavioral and psychological symptoms (such as hallucinations, delusions, irritability, and aggression) could markedly overload caregivers [12,15,16].

The average time duration for caring PWD ranges from 8 to 12 years [17]. Home is a major avenue for dementia care today. In Taiwan, 92.8% of dementia patients live in the community. They often rely on unpaid family caregivers such as spouses, daughters, and daughters-in-law [2]. Recent changes in social and family structures, declining birth rates, and rising employment of females, have increased the ratio of patients to caregivers. To cope with the expanding dementia population, the rising demand for long-term care, and the stressful events faced by the caregivers, it is particularly important to provide resources of care services apart from home care [18,19]. Sufficient social support to caregivers from the family or community and appropriate use of long-term care resources, e.g., community-based healthcare alternatives (respite care, home care, day care, supportive groups, etc.) are essential, because they could effectively reduce the burden and pressure of the caregivers, stabilize the emotional fluctuations, and help achieve equilibrium [4,9,20,21,22]. However, some studies reported no such benefits of resources [23,24,25]. In the case of PWD, caregivers are the key users of service resources. It could be an extremely important issue on how to provide various available services to meet the needs of both caregivers and the people under their care. Furthermore, the care services required for individual PWD varies due to the long-term course, diverse characteristics, and clinical manifestations of this disease. Therefore, it is an important issue to ensure care services are provided to meet the needs [26,27].

However, a number of studies reported that these services do not translate into practical benefits as much as expected [18,26,27]. According to the Taiwan Alzheimer’s Disease Association, more than half of PWD living at home do not use any formal services and are cared for solely by their families, with a low rate using home services at only 4.8% [2]. One study showed that 8% of the caregivers of dementia are long-term care service users, the main reason being that the caregivers do not perceive the needs for services (64%) [27]. Another study found more than 70% of these caregivers do not attend support group activities or use respite services. Most of them live with the patients and are their spouses. They tend to be older, more depressed, and have less social support. The extent of using available services is also related to the severity of the disease and its problematic behaviors [28]. Lim, et al. (2012) found that “knowing there is available service” increases the probability of using dementia care services. However, such increased uses by BPSD patients (*p* = 0.89) or care burdens (*p* = 0.47) did not reach statistical significance [18]. In short, no consistent results have been reported between use of care service resources and caregivers’ burden, social support, and BPSD patients. Clarifying the true relationships may enable the identification of useful strategies to improve the care of PWD, reducing the burden on caregivers.

Most studies have evaluated the severity of dementia patients based on the number and frequency of occurrences with BPSD. In addition to the frequency of BPSD, our present study further investigated which symptoms of BPSD are related to caregivers’ viewing of required care service resources. The aims of the study were three-folds: (a) to evaluate the utilization of long-term care service resources for caregivers of PWD; (b) to identify the relationship between the characteristics of patients and caregivers, the BPSD of care-recipients, caregivers’ burden, social support and the use of long-term care service resources; and (c) to explore the factors affecting caregivers in using long-term care service resources.

## 2. Materials and Methods

### 2.1. Study Design and Sampling

This is a cross-sectional study. Purposive sampling was applied, based on clinical diagnoses, to those patients with mild to moderate dementia. Clinical Dementia Rating (CDR) scores were between 1 and 2, and their family caregivers were serving at the neurologic and psychiatric outpatient departments of a medical center located at Xitum district (urban/moderate income) in central Taiwan. Based on physicians’ referrals, we distributed questionnaires to a total of 100 dyads (using the Fisher transformation method in our pilot study, we got a sample size of 97). The inclusion criteria were: main caregivers caring patients daily for more than 4 h, or were family decision-makers; caregivers were blood relatives or in-laws, aged 20 years or over; able to communicate in Mandarin or Taiwanese; and agreed to be interviewed. We excluded those in employment relationships. During the outpatient appointment, two trained researchers explained the study aim and contents of the questionnaire to both the patient and caregiver after the clinical examination. Data were collected by structured questionnaires and interviews of ~30 min.

### 2.2. Measures

To evaluate behavior and psychological symptoms of patients, participants were asked to complete the Dementia Behavior Disturbance Scale (DBD). To understand their care burden, social support network and the use of service resources, they were also asked to complete three additional questionnaires: Caregiver Burden Inventory (CBI), Functional Social Support Scale (FSS), and the Use of Long-Term Care Service Resources Scale. Questionnaires were translated from English to Chinese, with reliability and validity verified.

#### 2.2.1. Dementia Behavior Disturbance Scale (DBD)

The scale was designed by Baumgarten et al. in 1990 [29], consisting of 28 questions. The caregivers score the frequency of problem behavior during the pre-interview care period, ranging from 0 (never) to 3 (always). This scale has been applied to 96 dementia patients, and the Cronbach’s alpha is 0.83 [29]. The translated Chinese version (by Tang, 1991) [30], was used in a formal study with a Cronbach’s α of 0.87. The same α of this present study was 0.91, with a total score of 84. The higher the score, the higher is the frequency of the problem behaviors.

#### 2.2.2. Caregiver Burden Inventory (CBI)

Here we adopted the “Chinese Caregiver Burden Inventory, CCBI” from Kuo et al. (2014) [31], based on an original version developed by Novak and Guest (1989) [32]. The scale consists of 5 parts each containing a number of questions: (a) time burden of caring (the length of time caring for the patient), 5 questions; (b) burden of life development (life possibilities), 5 questions; (c) physical burden (feelings of physical conditions), 4 questions; (d) burden of social and family relationship (interpersonal relationship in family and social life), 4 questions; and (e) emotional burden (feelings for dementia patients), 4 questions. A 4-point Likert scale was used to code the answers of the total of 23 questions: 1: strongly disagree to 4: strongly agree. The possible total score ranged from 23 to 92. The higher the score, the heavier is the burden on the caregiver. The validity of this questionnaire (as translated by Kuo et al. 2014 into Chinese), was verified by construct validity and clinical validity. Its reliability, as verified by the internal consistency coefficient, showed a Cronbach’s α value of 0.95, and a split-half reliability of 0.88.

#### 2.2.3. Functional Social Support Scale (FSS)

This is based on the social support concept of Thoits (1982) [33] and translated into Chinese by Tang (1991) [30]. The 16 questions scale contains the following: 7 questions on instrumental support, providing material or behavioral support; 3 questions on informational support, providing guidance, advice, messages, or feedback information for problem solving; and 6 questions on emotional support, including care, love, and compassion. A 5-point Likert Scale was applied to answers, with scores from 0 to 4 points, and the cumulative score ranged from 0 to 64 points. The higher the score, the higher is the level of social support for the caregiver. Each dimension of internal consistency reliability (Cronbach’s α) was 0.88, 0.87, and 0.91 [30].

#### 2.2.4. The Use of Long-Term Care Service Resources Scale

Tailored to the aim of this study, a total of 19 care service resource items were listed according to the service resources of the long-term care policy announced by the Ministry of Health and Welfare [34]. They included mostly home-based (such as home services, home nursing, home-based rehabilitation, and reimbursement and rental of medical auxiliaries/equipment), community-based (such as daily care, dementia care, respite care, support group, transportation service, and temporary and short-term services), and institution-based services (such as nursing care, long-term care, and nursing home care). In the 4-point Likert scale, caregivers were scored by the frequency of use, based on a range from 0, for never; 1, seldom; 2, occasionally; and 3, usually used. The higher the score, the more frequently caregivers use them. In addition, the data were converted to binary categorical variables as 0 for “non-use” and 1 as “use”. Open questions were applied for answers by the participant and they had to indicate their reason for not using the resources, as well as the best way they would care for dementia patients.

#### 2.2.5. Control Variables

We recorded demographics data of PWD patients including gender, age, marital status, education level, religions, current living conditions, and other diseases if applicable. The demographic data of caregivers included gender, age, marital status, religion, education level, employment status, relationship with patients, living conditions (live or not with patient), number of years caring patients, co-caregivers, financial status, and self-rated health status.

### 2.3. Ethical Considerations

The study was approved by the Institutional Review Board (IRB) I and II of TVGH (No CE17114A; date of approval: 12 May 2017–11 May 2018). Participants all signed the informed consent.

### 2.4. Data Analyses

Descriptive statistics were applied to demographic data as well as the number, percentage, average, and standard deviation (SD) for each scale. The same wording as appeared in the open-ended questions was summarized and displayed with frequency and percentage. Inferential statistics including t-tests, chi-square tests, and ANOVA analyses were applied to assess differences between the demographic data/score of each questionnaire and the use of long-term care service resources. A Pearson’s correlation coefficient was used to determine the correlation between continuous variables and the frequency of use of long-term care resources. Logistic regressions on the use of long-term care resources and respective variables were interpreted using the odds ratio (OR). All statistical tests were two-tailed, and *p* < 0.05 was considered statistically significant. Data were analyzed with IBM SPSS software version 22.0.

## 3. Results

We collected a total of 100 questionnaires fully completed without missing information. For PWD patients, 52% were women vs 48% men, with an average age of 81.0 ± 9.8 years, mostly married (59%); the main education level was elementary school (55%) and 17% were illiterate, 45% of patients were Buddhist/Taoist, 83% were living with spouses and/or children or grandchildren, and 38% had other chronic diseases such as diabetes and hypertension (details shown in Table 1).

Table 1 shows that caregivers of PWD were mostly female (69%), with an average age of 54.3 ± 12.7 years and 22% were over 65 years; most of them were married (84%); education level was 46% college level or above; 43% were Buddhist/Taoist; nearly half of the caregivers holding an existing job; most caregivers were patients’ daughters (36%), followed by sons and spouses; 59% of caregivers were living with the patients; 68% had been caring the patients for over 2 years. On top of their caregiving, 61% of caregivers had other family members to help them, 73% of caregivers regarded their financial situation good. Self-rated health status was good in 24% of the caregivers, moderate in 47%, and poor in 29%.

The average care burden on the caregivers was 57.0 (SD = 12.4), with the highest score in time burden (M = 15.6, SD = 3.1), and the lowest score in emotional burden (M = 7.6, SD = 2.6). The social support score for caregivers was averaged at 41.3 (SD = 9.7), with the lowest score in the information support (M = 7.6, SD = 2.0) (Table 2).

Regarding the use of long-term care service resources, 40% of caregivers had never used them. Reasons for not using the resources were as follows: “not sure whether they fulfill the criteria to use those service resources” (38.2%), “never heard of the resources” (20.8%), and “not required” (19.4%). Regarding “the best way to care for dementia patients”, 62.5% caregivers responded, “care at home by family or others”, and only 17.3% caregivers responded “24-h care nursing home”.

The average frequency of BPSD in dementia patients was 28.4 (SD = 12.4, range 5 to 67). The top five behaviors responded in the questionnaire were: repeating the same theme; losing and misplacing things; sleeping a lot during the day; repeating the same action; and waking up at night for no reason. Analysis of variance showed that the use of long-term care service resources was significantly related with BPSD symptoms. These symptoms were hoarding for no reason, urinary incontinence, verbal or physical aggression, refusing others’ help with their hygiene, fecal incontinence, breaking out and getting lost, screaming for no reason (*p* < 0.05), anxious or restless, throwaway food, aggressive behavior, tearing up clothes, sexual harassment (*p* < 0.01) (Table 3).

Self-rated health status (regardless of living with patients), and BPSD of the PWD both showed statistically significant differences (*p* < 0.05) between caregivers using and not using long-term care service resources (Table 4).

Taking into account the frequency of using long-term care services resources, we found significant correlations (*p* < 0.05) between the age of the caregivers and BPSD of patients. That is, the younger the caregiver (Pearson correlation coefficient of −0.253), and the more frequent the BPSD (Pearson correlation coefficient 0.245), the higher was the frequency of using long-term care services resources (Table 5). In Table 5, BPSD of patients was significantly correlated (*p* < 0.01) with caregivers’ burden with a Pearson correlation coefficient of −0.464. That is, the more frequent the BPSD, the greater the caregivers’ burden. Burden on caregivers and social support of them were not related to the frequency of using long-term care service resources (*p* > 0.05).

Logistic regression analyses showed that long-term care service utilization was significantly affected by the frequency of patients’ BPSD and the living conditions of caregivers (Table 6). On average, increasing score on the frequency of BPSD by one point resulted in 1.045 times the probability of using long-term care service resources (OR = 1.045, *p* = 0.016, 95% CI = 1.008, 1.083), and 3.519 times the increase in the probability of using long-term care service resources while caregivers were not living with patients compared to living with patients (OR = 3.519, *p* = 0.007, 95% CI = 1.414, 8.759).

## 4. Discussion

We studied PWD caregivers in their use of long-term care service resources and their affecting factors. We found 40% of them not using care service resources. Particular care-recipients’ BPSD (such as urinary incontinence, verbal or physical aggression, refusing others’ help with their hygiene, fecal incontinence, breaking out and getting lost, screaming for no reason, anxious or restless, aggressive behavior, and sexual harassment), caregivers’ health status, and living with patients or not, affected significantly the caregivers‘ use of the care service resources. Furthermore, we found the frequency of BPSD and caregivers not living with patients increased the caregivers’ use of such care service resources.

Most caregivers are known to be unpaid women staying at home, typically spouses and daughters-in-law of the patients [1,6,10]. The present study showed the main caregiver for dementia was the daughter, followed by the son and spouse. Most of our caregivers were living with patients and had co-caregiver families nearby. This result is in line with the traditional Chinese family culture. In our study, 85% of caregivers had religious affiliation, and the care burden in Buddhist/Taoist caregivers was lower than those with other religious affiliations. Religions offer spiritual support for caregivers. As a result, they are spiritually satisfied and do not regard dementia care as a burden to them. Our results also showed that emotional burden has the lowest score regarding the care burden of caregivers. Such low scores are related to low motivation of caregivers in seeking care resources. Regarding “the best way to care for dementia patients”, 62.5% caregivers responded, “care at home by family or others”, and only 17.3% caregivers responded “24-h nursing home”. The reason for these choices in response could be related to, apart from high costs, some worries about criticism against them for not honoring their parents. We should point out that “filial piety” is a central value in traditional Chinese culture.

Our current study showed that caregivers not living with patients had a greater probability of using long-term care service resources than those living with patients. “The younger the caregiver, the higher the frequency of care resource utilization” indicated that younger caregivers are better able to search for information [35], and to actively search for available care resources to compensate for their lack of their full-time care and companionship due to work or other factors [18]. However, we found that the existing care service information was not properly accessed by caregivers. Therefore, accessibility of long-term care resources and the appropriateness of care knowledge and skills are important issues for the care professionals.

Caregivers of PWD are basically in a full-time engagement [1,2]. Not only are working hours long and unpaid, but also caregivers inevitably suffer from stress in the long run, leading to declines in their health status [13]. In the present study, we found that the status of the self-rated health of caregivers affected the use of care service resources. Other studies also reported health status of caregivers affecting their use of long-term care resources. The results are sending patients to institutions for care too soon, or to care centers unsuitable for PWD [11,13]. A number of studies also found that caregivers suffering from more than two chronic diseases show deteriorated health conditions within a year [3,18,36]. Our study showed similar findings: the self-rated health status of caregivers was 29% poor, and 47% moderate. However, it remains unclear if physical condition is directly related to the care of the PWD. This relationship can be further studied longitudinally to determine any causal effects. In any case, caregiving is a major health threat for caregivers, especially in the future when facing “elderly care for elderly”, health status of the caregivers is an unavoidable issue [7,9,10,36].

Previous studies have reported that BPSD is one of the most critical factors contributing to the care burden [7,9,10,15], especially agitation/aggression [15]. Our present study also found a significant relationship between the BPSD of patients, the caregivers’ burden, and their use of long-term care service resources. Certain symptoms of BPSD, such as urinary or fecal incontinence, physically or verbally aggressive behavior, when emerged, could embarrass caregivers more in their management, leading to their increased burden. Such situations could motivate their use of long-term care service resources. The problem behaviors (such as repeating the same theme, losing and misplacing things, sleeping a lot during the day, and repeating the same action), that are not too disturbing for caregivers might still induce them to search for related resources. Health professionals not only should help PWD families understand the disease and provide appropriate information, but also should deal with and treat BPSD case by case, especially when faced with distressful behaviors. In addition, listening to the voices and experiences of PWD and caregivers for direct assessment of actual behavior, the general population should be properly educated and assisted with propaganda. The aim is to reduce the likelihood of the dementia patients and their families in carrying along the stigma and hence getting negative feedback from the public [1,16,37]. Other studies showed that stigma increases burden on caregivers [2,38] and may affect their use of related resources.

We found that the low utilization of long-term care service resources by caregivers of PWD (particularly the long-term care services related to dementia) is likely due to the poverty of information available on related services. The caregivers were “not sure whether they fulfill the criteria” and “never heard of the resources,” demonstrating the importance of service information acquisition by the caregivers. The availability of resources affects the willingness of caregivers to utilize care resources [5]. Long-term care service resources should be embedded in the community, aiming at expanding user accessibility. We therefore, recommend that hospitals and communities should set up a “dementia integrating clinic” or dementia care teams. In addition to integrating the relevant professional departments, caregivers and patients on both sides should be viewed as a dyad, and equally provided with medical professional assessments, diagnosis, treatment, or transfer simultaneously, in order to maintain the caregivers’ health and to provide care-related knowledge and skills. The care team (e.g., physician, community and home care nurse) could also design the decision sharing model to help the caregivers choose the most appropriate long-term care service items to improve the care of PWD, reducing the burden on caregivers [39].

We found two important factors including patients’ BPSD and caregivers not living with the patient influence the long-term care service resources used, especially regarding certain problem behaviors and psychologic symptoms that could embarrass caregivers more and increase their burden. A larger/nationwide study for generalizability, and a stratified sampling database will be needed to strengthen our present results. This is a cross-sectional study, and caregivers had to recall the BPSD occurred in the past before filling out the questionnaire. Memory bias was unavoidable. A longitudinal research study in the future could help with data accuracy.

## 5. Conclusions

The study found BPSD, especially certain problem behaviors and psychologic symptoms, could more embarrass caregivers and increase their burden. Health professionals have to evaluate the patients’ problem symptoms and identify caregivers’ essential needs exactly. Appropriate individualized medical care should be provided and BPSD related long-term care service resources need to be referred to PWD and caregivers to take off their burden and improve the quality of care for patients with dementia.

## Figures and Tables

**Table 1 ijerph-17-06009-t001:** Demographic data of the study sample.

Variables		Mean (SD) or %	Caregivers
		Patients	
Gender	Male	48%	31%
	Female	52%	69%
Age		81.0 (9.8) (max 99 min 52)	54.3 (12.7) (max 81 min 22)
Marital status	Married	59%	84%
	Single	41%	16%
Education	Illiterate	17%	6%
	Elementary school	55%	7%
	High school	19%	41%
	College or above	9%	46%
Religion	Folk beliefs	29%	28%
	Buddhist/Taoist	45%	43%
	Christian/Catholic	13%	14%
	None	13%	15%
Living conditions	Living with family/patient	83%	59%
	Single or other	17%	41%
Relationship to the patients	Spouse		24%
	Son		28%
	Daughter		36%
	Daughter in law		8%
	Grandchild		5%
Employment status	Has a job currently (student)		47%
	Has no job or retired		40%
	Part-time job		13%
Number of years taking care of the patient	1 year or below		19%
	1–2 years		13%
	2 years or above		68%
Co-caregivers	No		25%
	Family member		61%
	Local/foreign worker		14%
Subjective financial status	difficult		18%
	still good		73%
	wellbeing		9%
Self-rated health status	poor		29%
	moderate		47%
	good		24%

**Table 2 ijerph-17-06009-t002:** Scores of Burden on caregivers and social support scales.

Variables	Min	Max	Mean	SD
**Caregiver Burden Scale**				
time burden	5	20	15.6	3.1
life development burden	5	20	12.8	3.5
physical burden	4	16	10.4	3.1
burden on social and family relationships	5	20	10.7	3.1
emotional burden	4	16	7.6	2.6
total score	24	91	57.0	12.4
**Functional Social Support Scale**				
instrumental support	7	28	18.2	4.7
informational support	3	12	7.6	2.0
emotional support	6	24	15.6	4.0
total score	16	64	41.3	9.7

**Table 3 ijerph-17-06009-t003:** Frequency ranking of psychological symptoms of dementia (BPSD) in patients with dementia (PWD) and differences between BPSD-related use and non-use of care resources.

Symptoms	Non-Use	Use	Weighting	Ranking	*p*
repeating the same question	2.18	2.03	211	1	0.34
losing things, misplace things	1.95	1.98	201	2	0.83
sleeping much during the day	1.83	1.67	179	3	0.35
repeating the same action	1.63	1.57	171	4	0.75
waking up at night for no reason	1.33	1.53	160	5	0.26
hoarding for no reason	1.05	1.47	156	6	0.05 *
other behaviors that bother you	1.15	1.00	152	7	0.54
anxious or restless	1.05	1.50	151	8	0.01 **
staying up at night walking or doing things	1.13	1.35	151	8	0.24
getting lost when going out alone	1.05	1.27	148	10	0.30
urinary incontinence	0.82	1.25	139	11	0.02 *
walking around at home or outdoors purposelessly	0.90	1.17	139	11	0.18
interest in daily activities	1.17	1.23	137	13	0.72
accusing others (e.g., of stealing their belongings)	0.77	1.07	135	14	0.14
verbal or physical aggression	0.87	1.18	133	15	0.02 *
eating too much	0.83	1.13	130	16	0.66
refusing others’ help with their hygiene	0.72	1.12	130	16	0.02 *
inappropriate dressing	1.03	0.95	129	18	0.67
refusing to eat	0.80	1.07	126	19	0.94
emotional incontinence, uncontrollable bursts out crying or laughing	0.80	0.85	121	20	0.76
fecal incontinence	0.45	0.82	121	20	0.03 *
breaking out and getting lost	0.30	0.67	117	22	0.03 *
throwaway food	0.23	0.67	116	23	0.01 **
aggressive behavior (e.g., hit, bite, catch, kick, spit, etc.)	0.15	0.58	111	24	0.00 **
tearing up clothes	0.13	0.48	111	24	0.01 **
indecently exposing private part of the body/ indecent behavior	0.20	0.42	109	26	0.10
sexual harassment	0.02	0.32	106	27	0.01 **
screaming for no reason	0.20	0.52	67	28	0.02 *

Note: * *p* < 0.05, ** *p* < 0.01.

**Table 4 ijerph-17-06009-t004:** The factors related to the difference of caregivers use or non-use of long-term care resources.

Variables		Non-Use	Use	
		Mean (SD) or %	Mean (SD) or %	*p*
Gender	Male	11	20	0.54
	Female	29	40	
Age		57.25 (13.36)	52.35 (12.99)	0.06
Marital status	single	4	12	0.18
	married	36	48	
Educational level	illiterate	3	3	0.80
	elementary school	3	4	
	high school	18	23	
	college or above	16	30	
Religion	fork beliefs	11	18	0.88
	Buddhist/Taoist	20	25	
	Christian/Catholic	5	8	
	None	4	9	
Employment status	has a job currently	15	31	0.08
	has no job or retired	21	19	
	part-time job	3	10	
	student	1	0	
Relationship to the patients	spouse	11	12	0.75
	son	10	18	
	daughter	14	22	
	daughter in law	4	4	
	grandchild	1	4	
Living conditions	live with patient	30	29	0.01 **
	not live with patient	10	31	
Number of years taking care of the patient	1 year or below	9	11	0.77
	1–2 years	4	9	
	2 years or above	28	40	
Co-caregivers	no	8	17	0.54
	family member	25	36	
	local/foreign worker	7	7	
Financial status	difficult common	6 29	12 44	0.55
	good	5	4	
Self-rated health status	poor	13	16	0.04 *
	moderate	13	34	
	good	14	10	
BPSD		24.63 (12.10)	30.85 (12.10)	0.02 *
Caregiver’s burden		54.35 (15.07)	58.75 (10.05)	0.08
Social support		40.39 (10.97)	41.92 (8.873)	0.61

Note: BPSD: Behavioral and Psychological Symptoms of Dementia, * *p* < 0.05, ** *p* < 0.01.

**Table 5 ijerph-17-06009-t005:** Factors correlated to the frequency of long-term care service resource utilization: Pearson’s correlation analysis.

Variables	Use of Resource	Caregiver’s Age	Patient’s BPSD	Caregiver’s Burden	Social Support
use of resource	1.000				
caregiver’s age	−0.253 *	1.000			
patient’s BPSD	0.245 *	−0.228 *	1.000		
caregiver’s burden	0.077	0.058	0.464 **	1.000	
social support	−0.003	−0.083	−0.421 **	−0.594 **	1.000

Note: BPSD: Behavioral and Psychological Symptoms of Dementia, * *p* < 0.05, ** *p* < 0.01.

**Table 6 ijerph-17-06009-t006:** Logistic regression model of long-term care service utilization by dementia caregivers.

Variables	B	S.E.	Wald	df	*P*	Exp(B)	95% CI for EXP(B)	
							Lower Limit	Upper Limit
constant	−1.288	0.584	4.856	1	0.028 *	0.276		
living condition	1.258	0.465	7.316	1	0.007 **	3.519	1.414	8.759
BPSD	0.044	0.018	5.800	1	0.016 *	1.045	1.008	1.083

Note: The reference group: living condition (caregivers not living with patients). BPSD: Behavioral and Psychological Symptoms of Dementia, * *p* < 0.05, ** *p* < 0.01.

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
