# Peer review of "The Needs and Utilization of Long-Term Care Service Resources by Dementia Family Caregivers and the Affecting Factors"

_ijerph, 2020, doi:10.3390/ijerph17166009_

Round 1

Reviewer 1 Report

  1. What is new for your study compared to others?
  2. It is hard to find your ultimate goal of your study. Need to clarify it.
  3. Do you have any reason to provide 100 sample size? it seems like you missed it in you paper.
  4. In my opinion, in your paper you applied structured questionnaires. But you mentioned you used semi-structured questionnaires. You should make it clear.
  5. You mentioned you will have DBD results but I am not able to see it. Need to provide it. It seems somehow critical.
  6. It is necessary to unify the decimal point as a whole. For example, Cronbach’s alpha coefficient is a mixture of two or three decimal places.
  7. At line 146-147 you provided the Cronbach’s alpha coefficient of FSS, I am not clear that this values are provided by your research or original study. Make it clear please.
  8. Based on your explanation, the measurements of long-term care service resources were nominal scale but you mentioned you have Likert scale in this manuscript.
  9. Your description on table 1 and 2 is not matched to the results.
  10. In Table 3, You have misprint on 117 for the ranking of ‘breaking out and get lost’
  11. In Table 4, Need to clarify the number of participants for each non-use and use group.
  12. In Table 6, is the p-value of the regression model .003? What is the value of r2 for explanatory power? Why did you select two independent variables for this case?
  13. Discussion has duplicated description that overlap with results. You need to modify (line 247-249, 307-309).
  14. You should discuss what you fore-mentioned research purposes. It seems you have discussed about general characteristics of participants. Better to modify your research purposes.
  15. In your discussion, you only displayed the limitations of your study. Need to describe what the strength of your study is?
  16. The conclusion is too general to describe what you have done in your study, better to be more specific on your study. And necessary to put what the specific implications derived from the results of your study?

Reviewer 2 Report

This seems to be a well designed and largely well presented study.  The English is for the most part OK, but with occasional mistakes in syntax and vocabulary.

I have comments on some details:

In the title, consider specifying caregivers as 'family caregivers'

Lines 43-44: ...prevalence rate of dementia in Taiwan is 12.39% by..... or every 5 in 100.  12.39% prevalence rate should mean 12.39 in every 100??? Please clarify.

52: often lead to ...........  Better would be: ..often suffer from physical stress leading to ailments such as............

79: ...of the participants....   What kind of participants, participating in what?  Please clarify

86:...uses by BPSD patients (p=0.031)...... did not reach statistical significance.  But p= 0.031<0.05, which is the most frequently applied level of significance, also in the present study. Misprint?

103: ...a medical center in central Taiwan.  To consider the generality and for comparison with other studies, more information shuld be given on the name and type of community in which the study was done. Give the name and some characteristics of the community. Urban/rural, low/high income..

119: [29]: should be [30]

121: This scale has been................0.83. Does this refer to a previous study? If so, a reference is needed

184-185: For PWD patients.......the highest education level was elementary school (38%)...........  According to Table 1, 19+9 percent of patients had high school or higher.  The information in text and Table should agree.

Table 1: The proportions of Married vs single is given as 59% vs 41% for patients and 84% vs 16% for caregivers.  The proportions Living with family vs single or other is given as almost exactly the inverse, 83% vs 17% for patients and 59% vs 41% for caregivers. One would expect a close correlation between being married and living with family, so are these numbers mixed up ?

194: Self-rated health status was good in 24%...........and poor in 29%. Table 1 shows the inverse, good in 29% and poor in 24%. Should agree

255-257: Most caregivers............typically spouses and daughters-in-law................ The present study confirmed a similar finding....  As it stands, the first sentence excludes sons and daughters, and the findings in the present study are not very similar
